# DISCOVERING THE MECHANICS OF HIDDEN NEURONS

## ABSTRACT

Neural networks trained through stochastic gradient descent (SGD) have been around for more than 30 years, but they still escape our understanding. This paper takes an experimental approach, with a divide-and-conquer strategy in mind: we start by studying what happens in single neurons. While being the core building block of deep neural networks, the way they encode information about the inputs and how such encodings emerge is still unknown. We report experiments providing strong evidence that hidden neurons behave like binary classifiers during training and testing. During training, analysis of the gradients reveals that a neuron separates two categories of inputs, which for layers close enough to the output remain impressively constant across training. During testing, we show that the fuzzy, binary partition described above embeds the core information used by the network for its prediction. These observations bring to light some of the core internal mechanics of deep neural networks, and have the potential to guide the next theoretical and practical developments.

## 1 INTRODUCTION

Deep neural networks are methods full of good surprises. Today, to perform image classification, one can train a 100M parameters convolutional neural network (CNN) with 1M training examples. Beyond raising questions about generalization (Zhang et al., 2017), it appears that the classification models derived from those CNNs offer object detectors for free, simply by thresholding activation maps (Yosinski et al., 2015; Zhou et al., 2015; Bau et al., 2017). The learned representations also appear to be universal enough to be re-used on new tasks even in an entirely different domain (e.g. from natural to medical images in Gulshan et al. (2016)). If memory or computation are bottlenecks, no problem, networks with binary weights and binary activations work just as well (Rastegari et al., 2016). What characteristics of SGD trained neural networks allow these intriguing behaviour to emerge?

Deep neural networks also have their limitations. They currently pose lots of difficulties with respect to continuous learning (Kemker et al., 2017), robustness (Szegedy et al., 2014; Nguyen et al., 2015), or unsupervised learning (Bojanowski & Joulin, 2017). Are there other good surprises to expect in those fields, or do those difficulties correspond to fundamental limitations of SGD trained deep neural networks?

In order to answer both questions, a better understanding of deep neural networks is definitely needed. Since the intricate nature of the network hinders theoretical developments, we believe experiments offer a valuable alternative path to offer an insight into the key mechanisms supporting the success of neural networks, thereby paving the way both for future theoretical and practical developments. In other words: analysing how something works helps understanding why it works, and gives ideas to make it work better.

In particular, the workings of hidden neurons, while being the core building block of deep neural networks, are still a mystery. It is tempting to associate hidden neurons to the detection of semantically relevant concepts. Accordingly, many works studying neurons have focused on their interpretability. A common and generally admitted conception consists in considering that they represent concepts with a level of abstraction that grows with the layer depth (LeCun et al., 2015). This conception has been supported by several works showing that intermediate feature maps in convolutional neural networks can be used to detect higher level objects through simple

thresholding ((Yosinski et al., 2015; Zhou et al., 2015; Bau et al., 2017). However, it is not clear if these observations reflect the entire relevant information captured by that feature map, or, on the contrary, if this interpretation is ignoring important aspects of it. In other words, the complete characterization of the way a neuron encodes information about the input remains unknown. Moreover, the dynamics of training that lead to the encoding of information used by a neuron is -to our knowledge- unexplored. This paper uses an experimental approach that advances the understanding of both these aspects of neurons. The main finding of our paper is the following: the encodings and dynamics of a neuron can approximately be characterized by the behaviour of a binary classifier. More precisely:

1. During training, we observe that the sign of the partial derivative of the loss with respect to the activation of a sample in a given neuron is impressively constant (except when the neuron is too far from the output layer). We observe experimentally that this leads a neuron to push activation of samples either up, or down, partitioning the inputs in two categories of nearly equal size.

2. During testing, quantization and binarization experiments show that the fuzzy, binary partition observed in point 1. embeds the core information used by the network for its predictions.

This surprisingly simple behaviour has been observed across different layers, different networks and at different problem scales (MNIST, CIFAR-10 and ImageNet). It seems like hidden neurons have a clearly defined behaviour that naturally emerges in neural networks trained with stochastic gradient descent. This behaviour has -to our knowledge- remained undiscovered until now, and raises intriguing questions to address in future investigations.

## 2 RELATED WORK

Previous works trying to understand the function of a neuron focus on its interpretability in terms of semantically relevant concepts. In the context of convolutional neural networks for image classification, several recent works have investigated how the activation of a single neuron is related to the input image, by developing methods to visualize the image structures that activate a neuron the most. Those methods include the training of a deconvolution network to project the feature activations back to the input pixel space (Zeiler & Fergus, 2014), and the analysis of how a neuron activation decreases when occluding portions of the input image, revealing which parts of the scene are important regarding this neuron activation (Zeiler & Fergus, 2014; Zhou et al., 2015). Inverse problem formulations have also been considered to reconstruct an image by inverting a representation obtained inside the network, using a gradient-descent approach regularized by different kinds of image models (Mahendran & Vedaldi, 2015; Yosinski et al., 2015). More recently, Bau et al. (2017) went a step further by developping methods to quantify the interpretability of the signal extracted through the previously described visualization methods. All those work conclude that (some of) the individual neurons have the capability to capture visually consistent structures. The fact that object detection emerges when considering units with highest activation inside a CNN trained to recognize scenes (Zhou et al., 2015) supports the idea that a binary form of encoding is embedded within the trained network. However, it is not clear if these observations reflect the entire relevant information captured by the studied feature map. Moreover, investigating further the emergence of concepts into neurons is also motivated by the observation that the object detection technique only works on a subset of the feature maps, leaving the understanding of the others as an open question. Our paper leaves interpretability behind, but provides experiments for the validation of a complete description of the encoding of information in any neuron.

Since the idea of binary encoding is central to our work, it is also related to works considering network binarization in a power consumption context, to mitigate the computational and memory requirements of convolutional network. In Courbariaux & David (2015), only the weights are constrained to only two possible values while, in Rastegari et al. (2016) and Hubara et al. (2016), both the filters and the inputs to convolutional layers are approximated with binary values. The fact

that those methods only induce a negligible loss in accuracy reveals that the conventional continuous definition of activations is certainly redundant. Motivated by those previous observations, our work further challenges the binary nature of individual neurons. It does not force binary activations during training, but instead reveals that a bimodal activation pattern naturally emerges from a conventional training procedure. While such an observation has already been presented in Agrawal et al. (2014) for ReLU networks, we go further by showing that there is no causal relation between the thresholding nature of the activation function and the binary encoding emerging in hidden neurons. Indeed, we show that a binary encoding emerges even in deep linear networks.

An important part of our work relies on the observation that the gradients used by the learning algorithm follow some consistent, predictable patterns. This observation has already been highlighted by Shwartz-Ziv & Tishby (2017) and Sinha et al. (2017). However, while these works focus on the gradients with respect to parameters on a batch of samples, we analyse the gradients with respect to activations on single samples. This difference of perspective is crucial for the understanding of the representation learned by a neuron, and is a key aspect of our paper.

## 3 PRELIMINARIES

Our goal is to describe the behaviour of neurons in a neural network. Given the growing complexity of neural networks, it is useful to define which part of the architecture we denote as a neuron. We associate neurons to activation functions: each application of a non-linear function to a single value defines one neuron. Following the literature, we will refer to the value preceding the application of the activation function as the pre-activation, and the result of it as the activation. In order to reflect the spatial structure of convolutional layers, we consider the different pixels of a feature map as different activations from a same neuron when studying statistical distributions.

We experiment with three different architectures: a 2-layer MLP with 0.5 dropout (Srivastava et al., 2014) trained on MNIST (LeCun et al., 1998), A 12-layer CNN with batchnorm (Ioffe & Szegedy, 2015) trained on CIFAR-10 (Krizhevsky & Hinton, 2009) and a 50-layer ResNet (He et al., 2016) trained on ImageNet (Deng et al., 2009). In addition to the ReLU activation (Nair & Hinton, 2010), we also analyse a version of the MLP with the sigmoid activation function, and a version of the 12-layer CNN without non-linear activation function. We will thus analyse five different models. Through the paper, we will repeatedly refer to specific layers of these networks. For the MLP, we simply refer to the two fully-connected layers as dense1-act and dense2-act, act being replaced by the used activation function (relu or sigmoid). The cifar CNN is divided in 4 stages of three layers. Layers from a stage have the same spatial dimensions and stages are separated by max-pooling layers. We refer to each layer through the index of its stage and the position of the layer inside the stage, starting at 0. Stage2layer0 refers thus to the first layer of the third stage. We use the ResNet50 network as provided by the Keras applications. We re-use their notations and refer to layers through their stage (in numbers) and block index (in letters). We only study the neurons after combination of the block outputs and the skip connections. The very first layer does not belong to a standard ResNet block, and is denoted as conv1. More information about the models and their training procedure can be found in Appendix. Our experiments were implemented using the Keras (Chollet & others, 2015) and Tensorflow (Agarwal et al., 2016) libraries.

## 4 NEURONS BEHAVE LIKE BINARY CLASSIFIERS DURING TRAINING

We start our quest of understanding a neuron by watching the gradients flowing through it. Most of the works analysing training dynamics of neural networks have focused on analysing gradients of the loss with respect to parameters, since these are directly used by the learning method. However, gradients with respect to the activations can also give us precious insights, since they directly reveal how the representation of a single sample is constructed.

## 4.1 THE REGULARITY OF ACTIVATION GRADIENTS

We proceed to a standard training of the cifar CNN and the MNIST MLP networks until convergence. During training, but in a separate process, we record the gradient of the loss with respect to the activations of each input on a regular basis (every 100 batches for cifar and every 10 batches for MNIST, leading to 1600 and 2350 recordings respectively). Measures were only performed on a random subset of neurons and samples due to memory limitations (see Appendix for more details). For each (input sample, neuron) pair, we compute the average sign of the partial derivatives with respect to the corresponding activation, as recorded at the different training steps. This value tells us whether an increased activation generally benefits (negative average) or penalizes (positive average) the classification of the sample. Due to the use of float32 precision, zero partial derivatives appear at some point in training when the sample is correctly classified, making the gradient very small. Since the signs of these values are not relevant, they are ignored when the average sign is calculated.

Figure 1 shows, for ten randomly selected neurons from different layers, the histograms of the computed average signs (there is one value per input sample). As one can see, the average partial derivative sign is either 1 or -1 for most of the samples, which indicates that the derivative sign doesn't change at all through the training. This is exactly the behaviour you would expect in the output of a binary classifier trying to separate two categories. Since around half of the activations have positive derivatives and the other half negative ones, a neuron seemingly tries to partition the input distribution in two distinct and nearly equally-sized categories. While training of neural networks could potentially be a very noisy procedure, we thus observe a remarkably clear and regular signal in the activation gradients. The regularity of training has already been observed for weights in Shwartz-Ziv & Tishby (2017) and Sinha et al. (2017), we observe it now through the lens of activations. In particular, we observe that the activation of a sample in a neuron should be pushed in the same direction throughout nearly all training to improve its prediction: either up or down. Histograms aggregating all neurons of a layer, can be found in Appendix.

This behaviour is much less apparent in layers far from the output. Indeed, the histogram corresponding to a neuron from stage2layer2-relu shows more sign changes than the one from stage3layer2-relu. Stage0Layer0-relu is even worse: the majority of the partial derivatives constantly change signs during training. This raises a question: are the same regular dynamics present in early layers, while hidden by undesirable noise? It has been observed that noise in gradients increases exponentially with depth in ReLU-networks due to the derivative discontinuity at 0 (Balduzzi et al., 2017). Indeed, the linear version of the cifar CNN (fourth row) provides a much clearer signal than the ReLU version (third row). However, other sources of noise are present: the histogram of stage0layer0-linear average derivative signs does not have a pronounced bimodal behaviour. Is the observed noise an inconvenience emerging from the architecture and training procedure, or rather a key aspect of learning? We leave this question as future work.

## 4.2 WATCHING NEURONS LEARN

The gradients strongly indicate that a neuron tries to separate two categories of inputs. Does this effectively happen during training? We assign each sample to a category based on its average activation partial derivative sign, and see how both categories' pre-activations evolve across the recordings. Categories are named 'low' and 'high' for positive and negative derivatives respectively. Figure 2 shows the results for a neuron in dense2-relu, dense2-sigmoid, stage3layer2-relu and in stage3layer2-linear. The dynamics of more neurons can be found in Appendix and in video format on the following link: https://www.youtube.com/channel/UC5VC2Oumb8r55sOkbNExB4A.

This visualization unveils a seemingly endless struggle to separate both categories. While very slow, the signal is effectively there: both categories are distinguished through the training procedure. However, training stops before both categories are completely separated. As will be discussed in Section 6, this raises a question that we believe is crucial: what mechanism regulates which samples are well partitioned in a neuron? To illustrate that the dynamics are not a simple translation, the final highest pre-activations are highlighted in yellow in the visualizations.

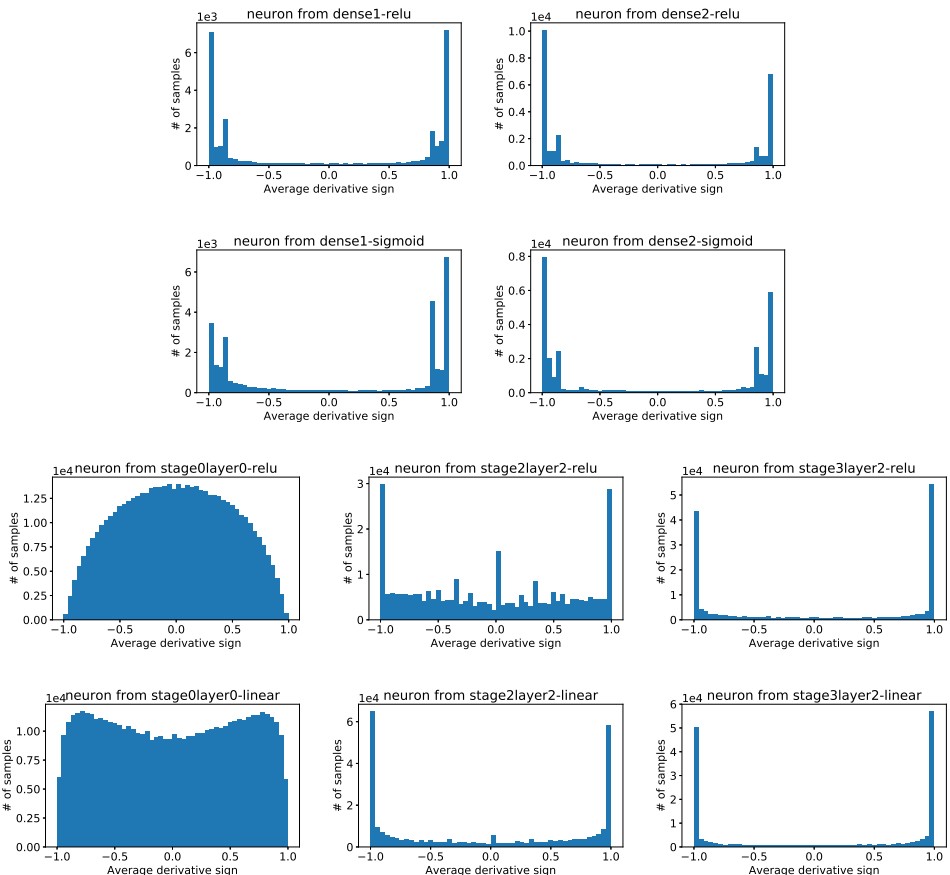

Figure 1: The figures show the histograms of the average sign of partial derivatives of the loss with respect to activation of samples, as collected over training for a random neuron in ten different layers. An average derivative sign of 1 means that the derivative of the activation of this sample was positive in all the recordings performed during training. For layers close enough to the output, we clearly observe two distinct categories: some sample activations should always go up, others always down. This reveals that the neuron receives consistent information about how to affect the activation of a sample, allowing it to act as a binary classifier. As detailed in Section 3, the layers from the first two rows are part of a network trained on MNIST (with ReLU and sigmoid activation functions respectively), the third and fourth row on CIFAR-10 (with ReLU and no activation function respectively).

Another question begs to be answered: according to which mechanism are the high and low categories defined? The average sign of the loss function partial derivative with respect to the activation of a sample determines the category, and seems to be constant along training -at least for layers close to the output (Figure 1). Categories are thus mainly fixed by the initialization of the network's parameters. Moreover, the sign of the derivative signal is heavily conditioned on the class of the input. In particular, in neurons of the output layer, partial derivative signs only depend on the class label, and not on the input. Figure 8 in Appendix shows that in dense2-relu, a class is in most cases either entirely present or absent of a category, and is only occasionally split across low and high categories. Category definition is thus approximately a selection of a random subset of classes, determined by the random initial parameters between the studied neuron and the output layer. We leave further exploration of these mechanisms as future work.

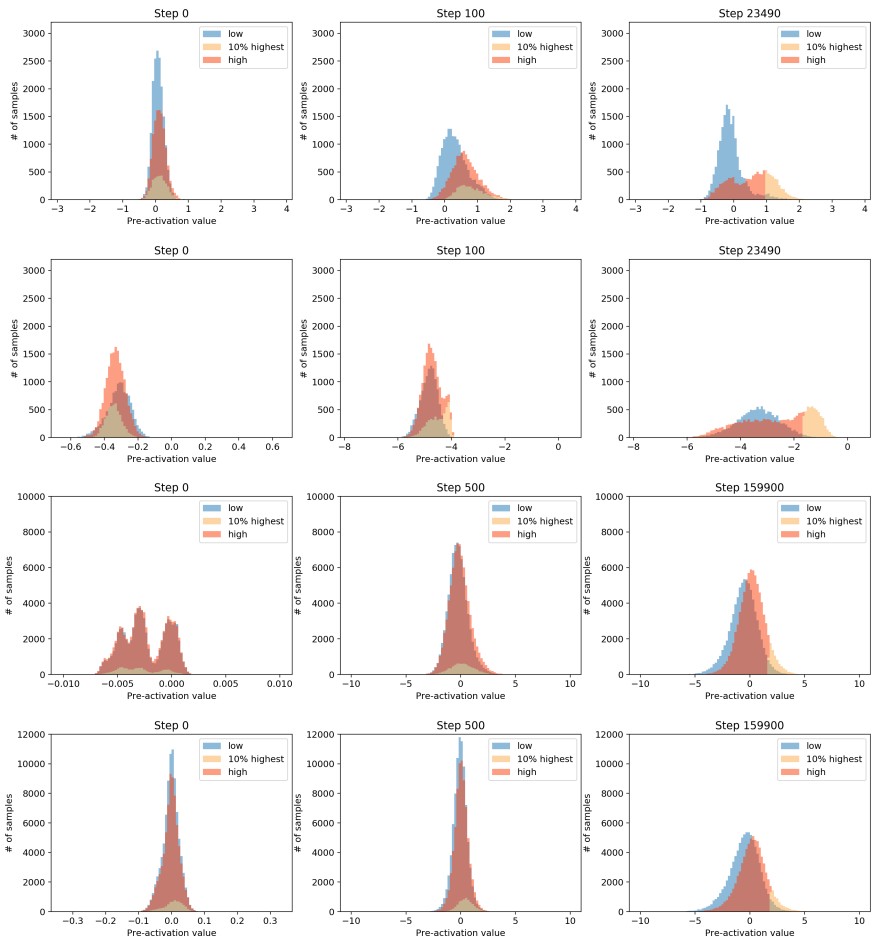

Figure 2: Evolution of the pre-activation distributions across training. Plots correspond to one neuron from dense2-relu (*first row*), dense2-sigmoid (*second row*), stage3layer2-relu (*third row*) and stage3layer2-linear (*fourth row*). Pre-activations are separated in two categories, high and low, based on the average partial derivative sign over training of their corresponding activation (see Figure 1). We can see that both categories are being separated during training. The final highest pre-activations of the high category are highlighted to show that it is not a simple translation. Supplementary images from other neurons can be found in Appendix and in video format on `https://www.youtube.com/channel/UC5VC2Oumb8r55sOkbNExB4A`.

## 5 THE BINARY BEHAVIOUR OF ACTIVATIONS

We have shown that neurons operate like binary classifiers during training. Does this also reflect the way a neuron encodes information about the input during testing? Even though the categories are not completely separated, does this partition provide the necessary information for the next layer? In this Section, we test if all the information a neuron transmits is encoded in the binary partition observed in the previous Section. We do this by studying how the performance of neural networks changes when activations of a trained layer are modified through specifically designed quantization and binarization strategies. The strategies are designed not only to highlight the hypothetical binary aspect of the encodings, but also to reveal structural components of it: how fuzzy is the binary rule and can we locate the thresholds? Moreover, this Section also studies ResNet50 since computational limitations are less of a problem.

## 5.1 A STUDY OF ROBUSTNESS TO PRE-ACTIVATION QUANTIZATION

The first experiment aims at testing if a neural network trained in a standard way is robust to quantization of pre-activations. Instead of accepting a continuous range of values, only two distinct values per neuron can be provided. Are these two values per neuron enough for transmitting the relevant information to the next layers? The quantization is based on the percentile rank of a pre-activations with respect to the pre-activation distribution of the neuron. For each neuron, percentiles are computed based on a subset of the data (training or test). The percentile corresponding to a chosen rank is then used as a threshold, separating the pre-activations in two distinct sets. A pre-activation will be quantized to the average value of the set it belongs to. Eleven thresholds equally spaced between 0 and 100 are tried out for the experiment. While the percentile is computed for each neuron specifically, the percentile rank used as a threshold is the same for all of them.

Figure 3 shows how accuracy on the test set is affected when quantization is performed on different layers. No form of training to adapt to this new pre-activation distribution is applied. The first and penultimate layers of each network are studied, as well as one intermediate layer for the cifar10 CNN and ResNet50. The signal is clear: neural networks are astonishingly robust to quantization of their pre-activations, although not explicitly designed to be so. Performance is quite robust to the chosen threshold, with a preference for higher percentile ranks. Amongst the 8 layers tested, only the conv1 layer from ResNet50 shows significant decrease in accuracy when its pre-activations are quantized. We believe this is due to poor the quality of the gradients in early layers, as discussed in Section 4.1.

## 5.2 A SLIDING WINDOW BINARIZATION EXPERIMENT

The quantization experiment suggests that each neuron transmits a binary signal to the next layer, which is a first step for confirming our hypothesis. But we still don't have a clear view on how the signal is encoded. Is there a clear threshold or a fuzzy rule? When can we be confident that a pre-activation should be considered as a member of the low category or the high one? What is the size of both categories? This section presents the design and results of a sliding window binarization experiment whose purpose is to provide insights around these questions.

In this experiment, instead of separating the pre-activations in two groups using a single percentile rank as threshold, we use two thresholds, forming a window. Activations between the two thresholds are mapped to 1, and activations outside of it are mapped to 0. The experiment is performed using a window with a width of 10 percentile ranks and a center that slides from rank 5 to rank 95. Thus, only 10% of all the pre-activations of a neuron are mapped to 1. Which 10% is fixed by the center of the window: if the center is at rank 35, only the activations between the 30 and 40 percentiles are mapped to 1. Similarly to the quantization experiment, the percentiles are computed using a randomly selected subset of the data (train or test).

With such a binarization method, the only information from the original signal that remains is if the activation was inside or outside the window. The usefulness of this information for a particular window depends on the coding scheme used by the neuron. The results can thus potentially provide insights about the organization of the representation and allow us to indirectly observe the binary partition used to encode information. To measure the usefulness of the transformed pre-activations, we monitor the test accuracy after reinitialization and retraining of the layers that follow the layer where the binarization has been performed. For computational reasons, linear classifier probes (Alain & Bengio, 2016) are used for analysing ResNet50 layers instead of retraining all the subsequent layers [1]. With this approach, we can verify if a network is able to make good use of the information contained in the binarized pre-activations and learn useful patterns that generalize to the test set. Since a neuron hypothetically transmits information through a binary partition of the inputs, performance of the network should be better when the pre-activations inside

---

[1]The original paper presenting linear probes mentions a problem emerging from the very high dimensionality of convolutional layers which is not suited for a linear classifier. To address this issue, we apply the linear probes on the global averages of the feature maps.

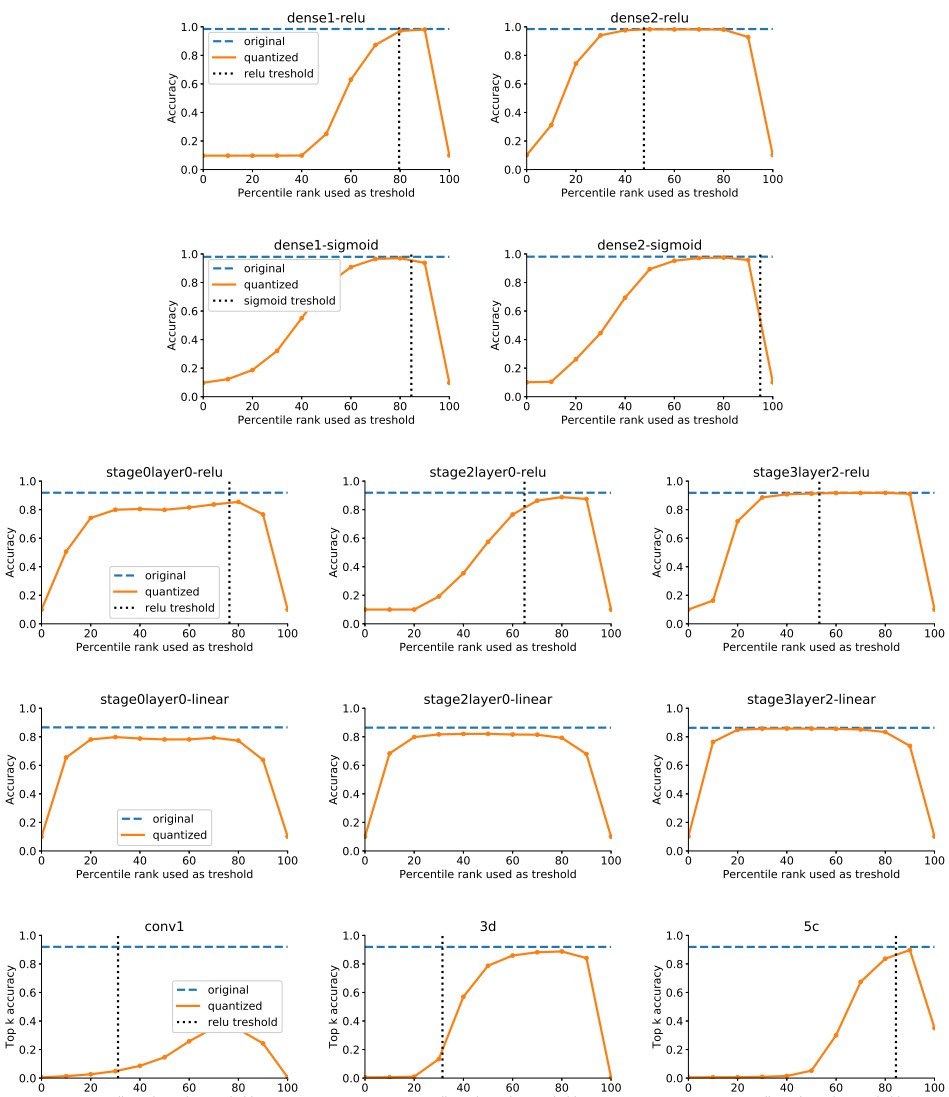

Figure 3: Quantization experiment: measuring test accuracy when pre-activations of a layer are quantized to two values per neuron, based on their percentile rank. Quantization is performed on a single layer at a time, using a range of percentile ranks as quantization thresholds. Except for conv1 (very first layer of ResNet50), the networks are astonishingly robust to quantization, suggesting that neurons provide a binary signal to the next layers. The average percentile rank of the zero pre-activation (which corresponds to ReLU's and sigmoid's threshold) is also provided. As detailed in Section 3, the layers from the first two rows are part of a network trained on MNIST (with ReLU and sigmoid activation functions respectively), the third and fourth row on CIFAR-10 (with ReLU and no activation function respectively) and the fifth row on ImageNet (with ReLU activation).

the window correspond to the same category. The performance should decrease when the window is located in fuzzy regions, where both categories are equally present. This experiment thus provide a tool to indirectly measure the presence of the two categories used by the coding scheme.

The results presented in Figure 4 show a clear signal across all layers and networks: the further away the center of the window is from rank 50, the better the performance of the network. Moreover, the symmetry around percentile rank 50 is striking. Given the binarization strategy, these results indicate a fuzzy partition of two categories, with a threshold around percentile rank 50, and a

confidence in the category that increases the higher (or lower) the activation. The fact that a window center at the 50th percentile rank does not induce random predictions indicates that the size of the categories are not always equal, but vary across neurons. Our results strengthen thus the hypothesis emerging from our analysis of the training dynamics, according to which a neuron partitions the inputs in two distinct but overlapping categories of quasi equal size. These new experiments tell us that this partition also characterizes how neurons encode information about the inputs.

Interestingly, there is no causal relation between the thresholding nature of activation functions and the binary behaviour that we observe in the pre-activations. Indeed, while the binary partition observed seems to be symmetrically arranged around the 50th percentile rank (Figure 4), the position of the ReLU or sigmoid thresholds (0 value) aren't (see Figure 3, or table 1 in Appendix). Moreover, the binary behaviour also emerges in linear networks, which don't have any thresholding effect in hidden neurons. This observation is quite unexpected, as previous studies on activation binarization focused on binarization at the threshold of the activation function (Agrawal et al., 2014), which now seems quite arbitrary.

## 6  DISCUSSION AND FUTURE WORK

In this paper, we try to validate an ambitious hypothesis describing the behaviour of a neuron in a neural network during training and testing. Our hypothesis is surprisingly simple: a neuron behaves like a binary classifier, separating two categories of inputs. The categories, of nearly equal size, are provided by the backpropagated gradients and are impressively consistent during training for layers close enough to the output. While stronger validation is needed, our current experiments, ran on networks of different depths and widths, all validate this behaviour.

Our results have direct implications on the interpretability of neurons. Studies analysing interpretability focused on the highest activations, e.g. above the 99.5 percentile in Bau et al. (2017). While these activations are the ones who are the most clearly discriminated by the neuron, we show that they do not reflect the complete behaviour of the neuron at all. Our experiments reveal that neurons tend to consistently learn concepts that distinguish half of the observed samples, which is fundamentally different.

We expect that our observations stimulate further investigations in a number of intriguing research directions disclosed by our analysis.

Firstly, since our analysis observes (in Fig.3 and 4) but does not explain the binary behaviour of neurons in the first layers of a very deep network, it would be interesting to investigate further the regularity of gradients (cfr. Section 4.1), in layers far from the output. This could potentially unveil simple training dynamics which are currently hidden by noise or, on the contrary, reveal that the unstable nature of the backpropagated gradients is a fundamental ingredient supporting the convergence of first layer neurons. Ultimately, these results would provide the missing link for a complete characterization of training dynamics in deep networks.

Secondly, our work offers a new perspective on the role of activation functions. Their current motivation is that adding non-linearities increases the expressivity of the network. This, however, does not explain why one particular non-linearity is better than another. Our lack of understanding of the role of activation functions heavily limits our ability to design them. Our results suggest a local and precise role for activation functions: promoting and facilitating the emergence of a binary encoding in neurons. This could be translated in activation functions with a forward pass consisting of well-positioned binarization thresholds, and a backward pass that takes into account how well a sample is partitioned locally, at the neuron level.

Finally, we believe that our work provides a new angle of attack for the puzzle of the generalization gap observed in Zhang et al. (2017). Indeed, combining our observations with the works on neuron interpretability tells us that a neuron, while not able to finish its partitioning before convergence, seems to prioritize samples with common patterns (cfr. Figure 2). This prioritization effect during training has already been observed indirectly in Arpit et al. (2017), and we are now

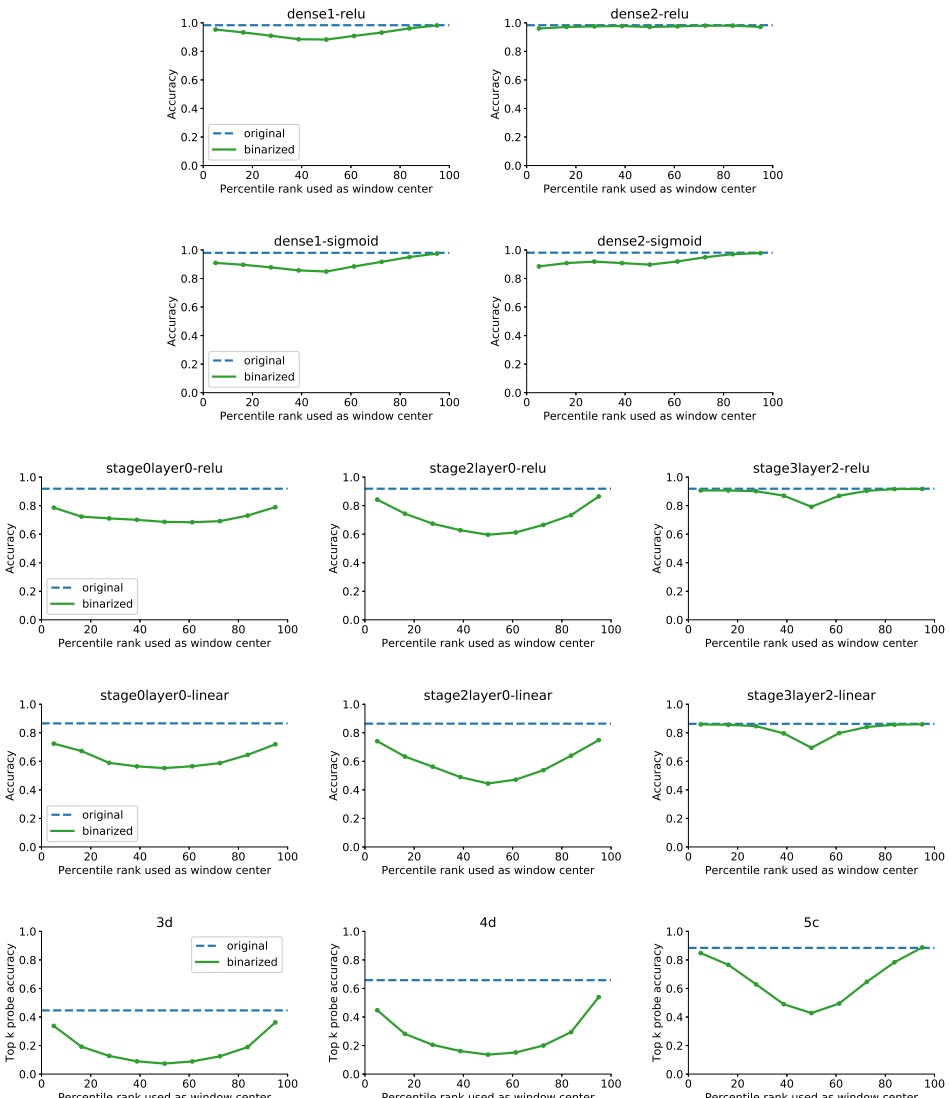

Figure 4: Sliding window binarization experiment: pre-activations inside a window with a width of percentile rank 10 are mapped to 1, pre-activations outside of it to 0. Information that remains in the signal is only the fact that the pre-activation was inside or outside the window. Observing if a new network can use this information for classification reveals structure about the encoding: which window positions provide the most important information for a classifier? The results show a clear pattern across all layers and networks that confirms an encoding based on a fuzzy, binary partition of the inputs in two categories of nearly equal size. As detailed in Section 3, the layers from the first two rows are part of a network trained on MNIST (with ReLU and sigmoid activation functions respectively), the third and fourth row on CIFAR-10 (with ReLU and no activation function respectively) and the fifth row on ImageNet (with ReLU activation).

able to localize and study it in depth. The dynamics behind this prioritization between samples of a same category should provide insights about the generalization puzzle. While most previous works have focused on the width of local minima (Keskar et al., 2017), the regularity of the gradients and the prioritization effect suggest that the slope leading to it also matters: local minima with good generalization abilities are stronger attractors and are reached more rapidly.

## 7 CONCLUSION

Two main lessons emerge from our original experimental investigation.

The first one arises from the observation that the sign of the loss function partial derivative with respect to the activation of a specific sample is constant along training for the neurons that are sufficiently close to the output, and states that those neurons simply aim at partitioning samples with positive/negative partial derivative sign.

The second one builds on two experiments that challenge the partitioning behaviour of neurons in all network layers, and concludes that, as long as it separates large and small pre-activations, a binarization of the neuron's pre-activations in an arbitrary layer preserves most of the information embedded in this layer about the network task.

As a main outcome, rather than supporting definitive conclusions, the unique observations made in our paper raise a number of intriguing and potentially very important questions about network learning capabilities. Those include questions related to the convergence of first layer neurons in presence of noisy/unstable partial derivatives, the design of activation functions, and the generalization puzzle.

ACKNOWLEDGMENTS

To be filled in.

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

# APPENDIX

## A INFORMATION ABOUT THE NETWORKS AND THEIR TRAINING PROCEDURE

### MNIST MLP

Architecture:
Dense (512) - Activation - Dropout (0.5) - Dense (512) - Activation - Dropout (0.5) - Dense (10)

Training information:

- Learning rate: $1e^{-1}$ for ReLU, 1 for sigmoid activation function
- Batch size: 128
- Number of epochs: 50

### CIFAR10 CNN

One layer is composed of a convolution, an activation function and BatchNormalization. We denote a layer with L(n) where n is the number of filters in the convolution. We denote maxpooling as MP and global average pooling as GP.

Architecture:
L(64) *3 - MP - L(128) *3 - MP - L(256)*3 - MP - L(512) *3 - GP - Dense(10)

Training information:

- Learning rate: $1e^{-2}$, divided by 5 after epoch 60
- Batch size: 32
- Number of epochs: 100
- Data augmentation is used (but not when retraining after binarization -Figure 4)

### RESNET50

This network is directly taken from the keras applications. Information about the architecture can be found at:
`https://github.com/fchollet/keras/blob/master/keras/applications/resnet50.py`

Training information is not provided.

## B  EXPERIMENT DETAILS

In Section 4, gradients and pre-activations are recorded for

- 30.000 samples for dense1 and dense2
- 10.000 for stage3layer2
- 5.000 for stage2layer2
- 500 for stage0layer0

The samples were randomly selected.

Computing percentiles is also performed on randomly selected samples for computation efficiency. More precisely, we limit the number of samples used to 100.000.

The probes used on ResNet50 used for Figure 4 used only 100.000 training samples from the ImageNet dataset. The test error, however, is computed on the complete ImageNet validation set.

## C  SUPPLEMENTARY IMAGES AND TABLES

Table 1: Average and standard deviation of the percentile rank of the ReLU threshold (0 value) across neurons of a layer. The percentile rank is based on the pre-activation distributions of each neuron. We observe that in the last layers, the position of the ReLU threshold is nearly the same for all neurons, suggesting convergence to a very precise position in the pre-activation distribution. Overall, given the noisy nature of its position, the ReLU threshold does not seem to be the cause of the binary behaviour of neurons observed in this paper.

| Layer | Average | Standard Deviation |
|---|---|---|
| dense1 | 80 | 8.9 |
| dense2 | 50 | 12 |
| stage0layer0 | 76 | 38 |
| stage2layer0 | 65 | 16 |
| stage3layer2 | 53 | **2.2** |
| conv0 | 31 | 37 |
| 3d | 31 | 20 |
| 5c | 84 | **2.8** |

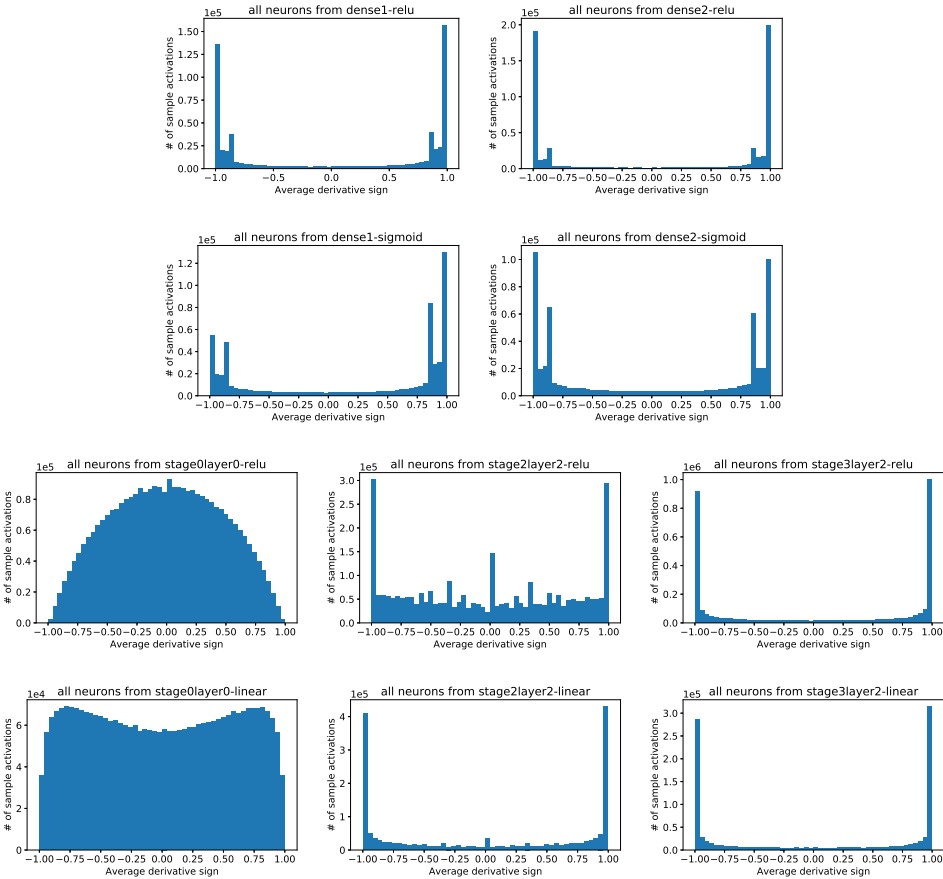

Figure 5: The figures show the histograms of the average sign of partial derivatives of the loss with respect to sample activations, as collected over training for all neurons in ten different layers. An average derivative sign of 1 means that the derivative of the activation of a neuron for this sample was positive in all the recordings performed during training. The histograms represent the statistics on all (neuron, sample) pairs of the layer. For layers close enough to the output, we clearly observe two distinct categories: some sample activations should always go up, others always down. This reveals that the neuron receives consistent information about how to affect the activation of a sample. The neuron-wise histograms in Figure 1 moreover show that more or less half of the input samples have negative derivatives, and the other positive ones, allowing a neuron to act as a binary classifier. As detailed in Section 3, the layers from the first two rows are part of a network trained on MNIST (with ReLU and sigmoid activation functions respectively), the third and fourth row on CIFAR-10 (with ReLU and no activation function respectively).

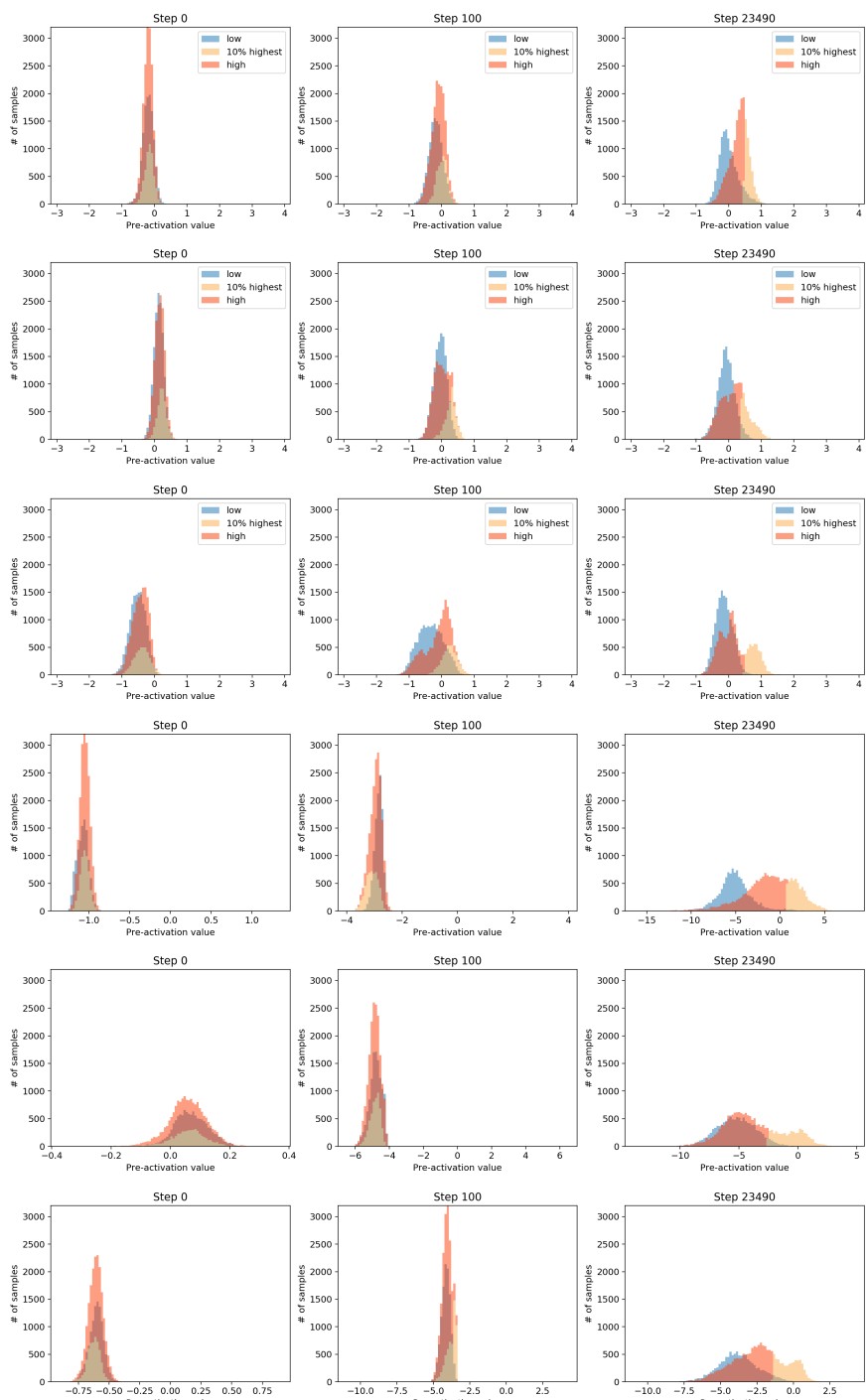

Figure 6: Evolution of the pre-activation distributions across training. Each line corresponds to the dynamics of a different neuron. Plots correspond to neurons from dense2-relu (*row 1-3*) and from dense2-sigmoid (*row 4-6*). Pre-activations are separated in two categories, high and low, based on the average partial derivative sign over training of their corresponding activation. We can see that both categories are being separated during training. The final highest pre-activations of the high category are highlighted to show that it is not a simple translation. These illustrations can be seen in video format on https://www.youtube.com/channel/UC5VC20umb8r55sOkbNExB4A.

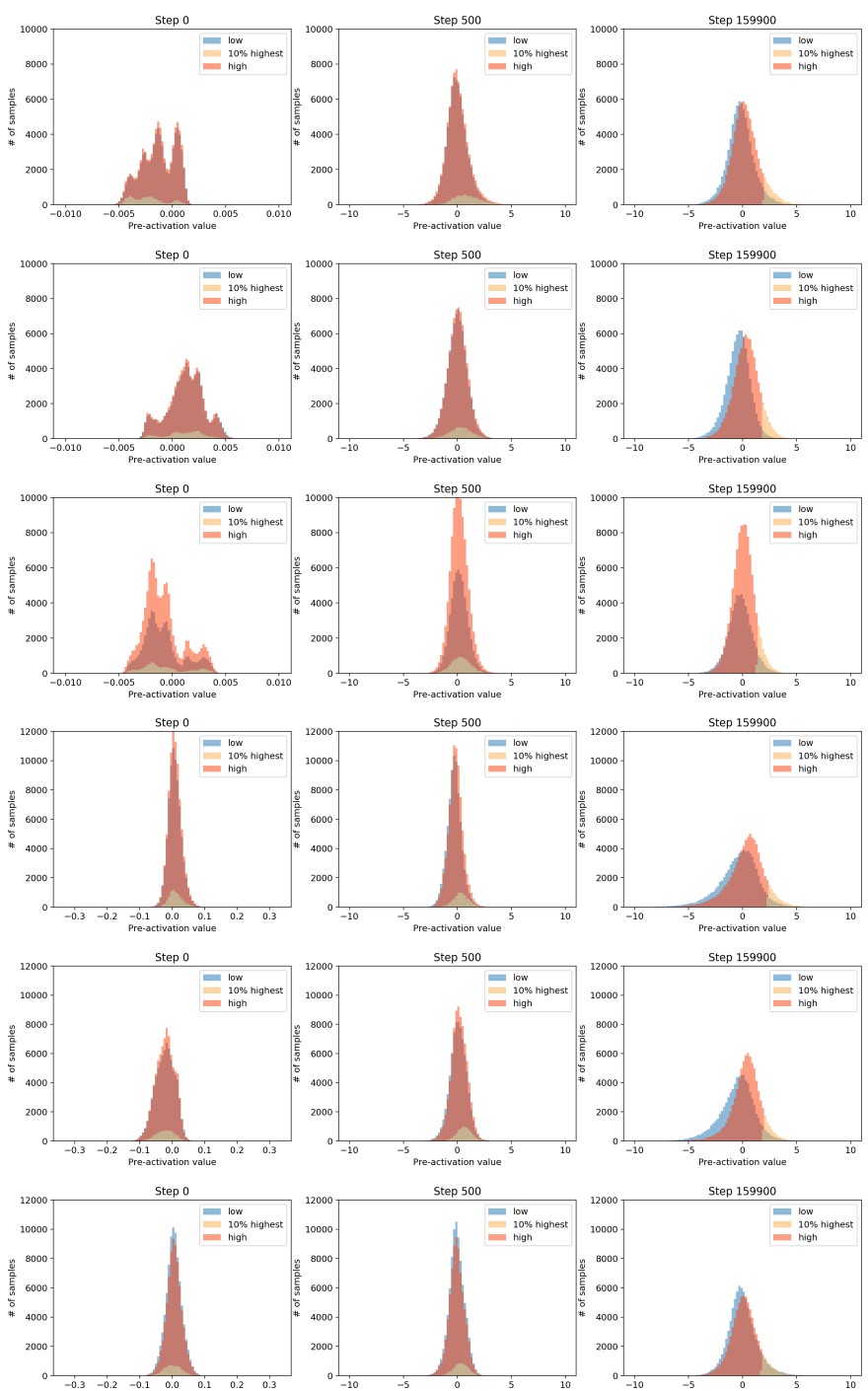

Figure 7: Evolution of the pre-activation distributions across training. Each line corresponds to the dynamics of a different neuron. Plots correspond to neurons from stage3layer2-relu (*row 1-3*) and from stage3layer2-linear (*row 4-6*). Each line corresponds to the dynamics of a different neuron. Pre-activations are separated in two categories, high and low, based on the average partial derivative sign over training of their corresponding activation. We can see that both categories are being separated during training. The final highest pre-activations of the high category are highlighted to show that it is not a simple translation. These illustrations can be seen in video format on https://www.youtube.com/channel/UC5VC20umb8r55sOkbNExB4A.

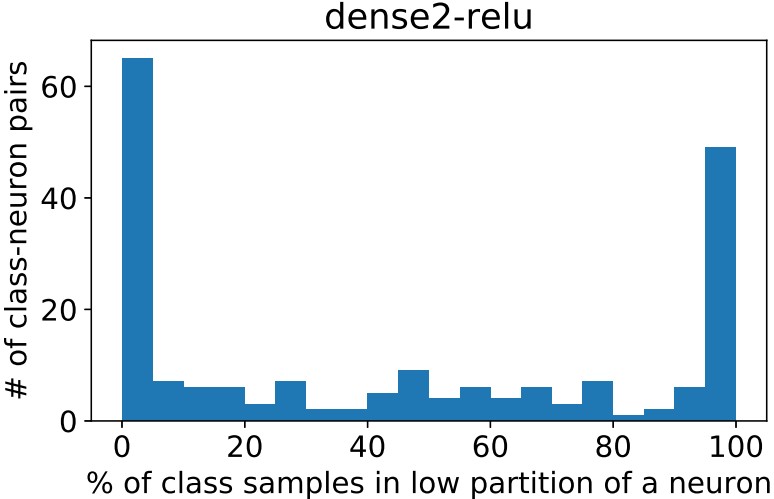

Figure 8: Histogram showing the consistency between the class of a sample and its belonging to the low category of a neuron (samples whose activation should be decreased) in dense2-relu. In most cases, nearly all the elements of a class are in the same category, as can be seen by the two peaks at 0 and 100%.

