# OpenReview forum: "Discovering the mechanics of hidden neurons"
_ICLR.cc/2018/Conference — Reject_

### Official Review · AnonReviewer2 · 2017-11-26
**interesting finding but lacking an explanation**

**Rating:** 4
**Confidence:** 4

**Review:**

The paper proposes to study the behavior of activations during training and testing to shed more light onto the inner workings of neural networks. This is an important area and findings in this paper are interesting!

However, I believe the results are preliminary and the paper lacks an adequate explanation/hypothesis for the observed phenomenon either via a theoretical work or empirical experiments.
- Could we look at the two distributions of inputs that each neuron tries to separate?
- Could we perform more extensive empirical study to substantiate the phenomenon here? Under which conditions do neurons behave like binary classifiers? (How are network width/depth, activation functions affect the results).

Also, a binarization experiment (and finding) similar to the one in this paper has been done here:
[1] Argawal et al. Analyzing the Performance of Multilayer Neural Networks for Object Recognition. 2014

+ Clarity: The paper is easy to read. A few minor presentation issues:
- ReLu --> ReLU

+ Originality:
The paper is incremental work upon previous research (Tishby et al. 2017; Argawal et al 2014).

+ Significance:
While the results are interesting, the contribution is not significant as the paper misses an important explanation for the phenomenon. I'm not sure what key insights can be taken away from this.

---

> ### Author Response · Authors · 2017-12-18
> **part 3**
>
>
> ------
> + Clarity: The paper is easy to read. A few minor presentation issues:
> - ReLu --> ReLU
> ------
>
> Thanks for noticing it! It is now changed.
>
> ------
> + Originality:
> The paper is incremental work upon previous research (Tishby et al. 2017; Argawal et al 2014).
> ------
>
> Above, we have already commented on our contribution in the light of Argawal et al. We hope that it makes clear that our paper provides original work compared to them.
> On the other hand, the only common point between Tishby et al. and our work lies in the fact that both works analyze the regularity of gradients during training. However, like our paper specifies, “while these works (including Tishby et al.) focus on the gradients with respect to parameters on a batch of samples, we analyze the gradients with respect to activations on single samples. This difference of perspective is crucial for the understanding of the representation learned by a neuron, and is a key aspect of our paper.” With Tishby et al.’s results, it is impossible to make a link between hidden neurons and binary classification of individual samples, which is the core observation of our paper.
>
> ------
> + Significance:
> While the results are interesting, the contribution is not significant as the paper misses an important explanation for the phenomenon. I'm not sure what key insights can be taken away from this.
> ------
>
> We agree with you that our paper lacks a final polished and complete conclusion. Indeed, we don’t see our paper as finished work, but rather as the opening of a promising investigation direction for a problem that has remained unsolved for more than 30 years: understanding neural networks. The fact that our observations are not obvious and generalize over very different networks suggests that these are very important properties to know in order to understand neural networks. The fact that the design and intuitions behind our experiments are not trivial and presenting them is already a challenge makes us believe it deserves to be presented to the community and discussed in an interactive manner. To emphasize the research directions that emerge from our observations, we have updated the ‘Discussion and future work’ section of our paper. In particular, we describe three directions of research related to the training dynamics of layers far from the output, the design of activation functions, and the generalization puzzle.

---

> ### Author Response · Authors · 2017-12-18
> **part 2**
>
>
> ------
> Also, a binarization experiment (and finding) similar to the one in this paper has been done here:
> [1] Argawal et al. Analyzing the Performance of Multilayer Neural Networks for Object Recognition. 2014
> ------
>
> Thanks for pointing out this reference.
> We were not aware of it and will add it in the related work section of our paper. We however consider that our paper brings contributions and makes observations that are different or that significantly complement the ones made by Argawal et al. Argawal et al. analyze the properties of features/activations in a transfer learning framework. In relation with our contribution, Section 5.1 of their paper shows that the binarization of the activation at ReLU threshold (at 0) doesn’t hurt performance on the new task. The claims of our paper go way beyond this observation. We hope that the new version of our paper makes them clearer. Two main differences are as follows:
>
> - Argawal et al. binarize the activations in a very simple manner: according to the threshold of the activation function. In our paper, we show that they missed a key insight: the binary behaviour of features/activation is not related to the thresholding nature of activation functions. The reality is much more subtle: the binary behaviour is deeply linked with the SGD training dynamics of deep networks (whatever their activation function), and the partition threshold systematically lies around 50 percentile rank (and not around the arbitrary zero ReLU threshold). This observation about the partition threshold directly emerges from the comparison between the clear pattern of Figure 4 and the much noisier position of ReLU thresholds (Figure 3 or Table 1). This claim is now made even stronger by our experiments with a linear CNN, without any ReLU (or activation function related) thresholding. Our observations are thus much stronger and unexpected than the one from Argawal et al.
>
> - The dynamics of training are not explored at all in Argawal et al. Our paper shows, for the first time, that the dynamics in neurons that are close enough to the output adopt the ones from binary classifiers. While restricted to a subset of layers (i.e. the ones that are sufficiently close to the output), the simplicity of the dynamics is unexpected, and even appears in MLP networks that have been around for more than 30 years. Moreover, it is possible that similar behaviour appears in early layers, but is hidden by unnecessary noise in the backpropagated gradients. Verifying this hypothesis requires further investigations, but our work makes a first step towards a broader characterization of training dynamics.
>
> We’ve added a short discussion on the relation of our work with Argawal et al. in the Related work section (Section 2, end of second paragraph) which is further discussed in the last paragraph of section 5.2.

---

> ### Author Response · Authors · 2017-12-18
> **Thank you for your comments. Here's part 1 of our answer.**
>
> Thank you for your comments. We’ve made our best to account for them in the revised version of our paper. Below, we present answers to your specific comments. Moreover, let us bring to your attention that changes have been made in section 4.1, clarifying greatly the experimental approach we used. More information about it can be found in our answers to reviewer 3.
>
> ------
> - Could we look at the two distributions of inputs that each neuron tries to separate?
> ------
>
> Since the distributions are determined by the sign of the loss function partial derivative (with respect to the neuron activation), the two distributions of inputs in one neuron are currently only available in layers close enough to the output layer, where the partial derivative sign remains constant along training. For such layers, we can get intuitions about the content of the distributions through the following reasoning (added as last paragraph of section 4.2):
> The average sign of the loss function partial derivative with respect to the activation of a sample determines the category, and seems to be constant along training -at least for layers close to the output (Figure 1). Categories are thus mainly fixed by the initialization of the network's parameters. Moreover, the sign of the derivative signal is heavily conditioned on the class of the input. In particular, in neurons of the output layer, partial derivative signs only depend on the class label, and not on the input. Figure 8 in appendix shows that in dense2-relu, a class is in most cases either entirely present or absent of a category, and is only occasionally split across low and high categories. Category definition is thus approximately a selection of a random subset of classes, determined by the random initial parameters between the studied neuron and the output layer.
>
> ------
> - Could we perform more extensive empirical study to substantiate the phenomenon here? Under which conditions do neurons behave like binary classifiers? (How are network width/depth, activation functions affect the results).
> ------
>
> Thank you for this suggestion.
> To complete our empirical study, we have considered other activations than the ReLU one.  Networks with sigmoid and linear activations are now considered by our analysis (see Figures 1, 2, 3 and 4). As we expected, the results are the same, even with purely linear networks. This emphasizes a message that was not well enough explained in the original paper: the observations we make are not caused by the thresholding behaviour of activation functions (ReLU, sigmoid), but are deeply linked with the training dynamics of deep neural networks. This observation is now discussed in the last paragraph of section 5.2.
> Regarding the impact of the network architecture (width, depth, connectivity), note that it is reasonably explored in the original paper: the same conclusions emerge from a 512-wide two-layer MLP, from a 12-layer CNN with a width of up to 512 filters, and from a 50-layer ResNet with a width of up to 2048 filters.

---

### Official Review · AnonReviewer1 · 2017-11-27
**Experimental study on how units of CNNs behave as binary classifiers**

**Rating:** 5
**Confidence:** 4

**Review:**

This paper presents an experimental study on the behavior of the units of neural networks. In particular, authors aim to show that units behave as binary classifiers during training and testing.

I found the paper unnecessarily longer than the suggested 8 pages. The focus of the paper is confusing: while the introduction discusses about works on CNN model interpretability, the rest of the paper is focused on showing that each unit behaves consistently as a binary classifier, without analyzing anything in relation to interpretability.  I think some formal formulation and specific examples on the relevance of the partial derivative of the loss with respect to the activation of a unit will help to understand better the main idea of the paper. Also, quantitative figures would be useful to get the big picture. For example in Figures 1 and 2 the authors show the behavior of some specific units as examples, but it would be nice to see a graph showing quantitatively the behavior of all the units at each layer. It would be also useful to see a comparison of different CNNs and see how the observation holds more or less depending on the performance of the network.

---

> ### Author Response · Authors · 2017-12-18
> **Sorry for lack of clarity in original version. We've found what was causing confusion and revised it.**
>
> We are sorry that the original version did not allow you to fully understand the main ideas of the paper. Thanks to the reviews and comments, we’ve noticed that indeed, some parts were not explained clearly enough, and we have done our best to clarify these in the new version.
>
> The link with interpretability of neurons is that both works try to understand the role of a neuron inside a neural network. Our approach, however, is different. As stated in the related work section, “our paper leaves interpretability behind, but provides experiments for the validation of a complete description of the encoding of information in any neuron.” Our discussion and future work section emphasizes the impact of our observations on neuron interpretability.
>
> We modified the text to make it explicit when the activation of a single sample is considered (in contrast to the average on a mini-batch). This implies replacing ‘the partial derivative of the loss with respect to the activation’ by ‘the partial derivative of the loss with respect to the activation OF ONE SAMPLE’. While clear in our mind, we now noticed that our initial phrasing was confusing (see also our answer to reviewer 3). We hope that the new version of section 4.1 makes the relevance of the recorded partial derivatives more clear.
>
> We’ve added quantitative figures aggregating all neurons of a layer. It reveals that the aggregate behavior follows the same pattern than the examples provided in Figure 1. Due to space limitations, we’ve added the new figures to the appendix. Finally, we’ve also added experiments with sigmoid activation function and purely linear networks, revealing the same behaviour.
>
> We’ve made efforts to reduce the length of the paper (i.e. removing section about ReLU analysis). However, due to the addition of new figures and comments requested by the reviewers, the number of pages has increased in the new version. We believe reducing the length of the paper would penalize its clarity. However, we will account for the reviewer’s opinion if it is maintained.

---

### Official Review · AnonReviewer3 · 2017-11-29
**Potentially amazing results obscured by poor (but fixable!) explanation**

**Rating:** 7
**Confidence:** 4

**Review:**

--------------------
Review updates:
Rating 6 -> 7
Confidence 2 -> 4

The rebuttal and update addressed a number of my concerns, cleared up confusing sections, and moved the paper materially closer to being publication-worthy, thus I’ve increased my score.
--------------------

I want to love this paper. The results seem like they may be very important. However, a few parts were poorly explained, which led to this reviewer being unable to follow some of the jumps from experimental results to their conclusions. I would like to be able to give this paper the higher score it may deserve, but some parts first need to be further explained.

Unfortunately, the largest single confusion I had is on the first, most basic set of gradient results of section 4.1. Without understanding this first result, it’s difficult to decide to what extent the rest of the paper’s results are to be believed.

Fig 1 shows “the histograms of the average sign of partial derivatives of the loss with respect to activations, as collected over training for a random neuron in five different layers.” Let’s consider the top-left subplot of Fig 1, showing a heavily bimodal distribution (modes near -1 and +1.). Is this plot made using data from a single neuron or from  multiple neurons? For now let’s assume it is for a single neuron, as the caption and text in 4.1 seem to suggest. If it is for a single neuron, then that neuron will have, for a single input example, a single scalar activation value and a single scalar gradient value. The sign of the gradient will either be +1 or -1. If we compute the sign for each input example and then AGGREGATE over all training examples seen by this neuron over the course of training (or a subset for computational reasons), this will give us a list of signs. Let’s collect these signs into a long list: [+1, +1, +1, -1, +1, +1, …]. Now what do we do with this list? As far as I can tell, we can either average it (giving, say, .85 if the list has far more +1 values than -1 values) OR we can show a histogram of the list, which would just be two bars at -1 and +1. But we can’t do both, indicating that some assumption above was incorrect. Which assumption in reading the text was incorrect?

Further in this direction, Section 4.1 claims “Zero partial derivatives are ignored to make the signal more clear.” Are these zero partial derivatives of the post-relu or pre-relu? The text (Sec 3) points to activations as being post-relu, but in this case zero-gradients should be a very small set (only occuring if all neurons on the next layer had either zero pre-relu gradients, which is common for individual neurons but, I would think, not for all at once). Or does this mean the pre-relu gradient is zero, e.g. the common case where the gradient is zeroed because the pre-activation was negative and the relu at that point has zero slope? In this case we would be excluding a large set (about half!) of the gradient values, and it didn’t seem from the context in the paper that this would be desirable.

It would be great if the above could be addressed. Below are some less important comments.

Sec 5.1: great results!

Fig 3: This figure studies “the first and last layers of each network”. Is the last layer really the last linear layer, the one followed by a softmax? In this case there is no relu and the 0 pre-activation is not meaningful (softmax is shift invariant). Or is the layer shown (e.g. “stage3layer2”) the penultimate layer? Minor: in this figure, it would be great if the plots could be labeled with which networks/datasets they are from.

Sec 5.2 states “neuron partitions the inputs in two distinct but overlapping categories of quasi equal size.” This experiment only shows that this is true in aggregate, not for specific neurons? I.e. the partition percent for each neuron could be sampled from U(45, 55) or from U(10, 90) and this experiment would not tell us which, correct? Perhaps this statement could be qualified.

Table 1: “52th percentile vs actual 53 percentile shown”.

> Table 1: The more fuzzy, the higher the percentile rank of the threshold

This is true for the CIFAR net but the opposite is true for ResNet, right?

---

> ### Author Response · Authors · 2017-12-18
> **Thank you very much for these encouraging and involved comments. Here is part 1 of our answer.**
>
> Thank you very much for these encouraging and involved comments. We’ve done our best to answer them appropriately, and are looking forward to your feedback.
>
> --------
> Fig 1 shows “the histograms of the average sign of partial derivatives of the loss with respect to activations, as collected over training for a random neuron in five different layers.” Let’s consider the top-left subplot of Fig 1, showing a heavily bimodal distribution (modes near -1 and +1.)....
> ... As far as I can tell, we can either average it (giving, say, .85 if the list has far more +1 values than -1 values) OR we can show a histogram of the list, which would just be two bars at -1 and +1. But we can’t do both, indicating that some assumption above was incorrect. Which assumption in reading the text was incorrect?
> --------
>
> Thanks for your comment, which reveals a lack of clarity in our explanation. When analyzing the derivatives, we treat two dimensions separately: input samples and training step. When recording the partial derivatives of an activation, we keep track of both dimensions, such that we can easily access the derivative signs of the activation of a single sample across the training procedure. To create the histograms of figure 1, we first compute, for each individual sample separately, the average of the derivative signs over all the recorded training steps. This tells us whether an increased activation generally benefits (negative average) or penalizes (positive average) the classification of this sample. To extend the analysis to all samples, the histogram of average signs of derivatives (one scalar per sample) is plotted over all input samples.
>
> When reading the manuscript at the light of your comment, we have observed that the confusion is largely induced by the fact that we generally use the term ‘activation’ to refer to the ‘activation of a single sample’. Example:
> What we have written: “In particular, we observe that an activation is pushed in the same direction throughout nearly all the training: either up or down”
> What we had in mind: “In particular, we observe that the activation OF A SAMPLE is pushed in the same direction throughout nearly all the training: either up or down”
> Similarly, when talking about “the average sign of partial derivatives with respect to an activation”, we had in mind “the average sign of partial derivatives with respect the activation of a sample”
> The revised version is much clearer in this regard.
>
> --------
> Further in this direction, Section 4.1 claims “Zero partial derivatives are ignored to make the signal more clear.” Are these zero partial derivatives of the post-relu or pre-relu? The text (Sec 3) points to activations as being post-relu, but in this case zero-gradients should be a very small set (only occuring if all neurons on the next layer had either zero pre-relu gradients, which is common for individual neurons but, I would think, not for all at once). Or does this mean the pre-relu gradient is zero, e.g. the common case where the gradient is zeroed because the pre-activation was negative and the relu at that point has zero slope? In this case we would be excluding a large set (about half!) of the gradient values, and it didn’t seem from the context in the paper that this would be desirable.
> --------
>
> We indeed analyze post-relu derivatives. Zero derivatives actually emerge for a sample when it is well classified, making the gradients too small to be handled by the float32 precision (smallest number is 1.19209e-07). Since the notion of sign is not relevant anymore for zero values, we compute the average of partial derivative signs for a sample only over the training steps for which the partial derivative is non-zero. We have made the reasoning explicit in the paper! Thanks for pointing it out.
>
> In particular, the first paragraph of section 4.1 has been revised to account for your first two questions:
> “We proceed to a standard training of the cifar CNN and the MNIST MLP networks until convergence. During training, but in a separate process, we record the gradient of the loss with respect to the activations of each input on a regular basis (every 100 batches for cifar and every 10 batches for MNIST, leading to 1600 and 2350 recordings respectively). Measures were only performed on a random subset of neurons and samples due to memory limitations (see Appendix for more details). For each (input sample, neuron) pair, we compute the average sign of the partial derivatives with respect to the corresponding activation, as recorded at the different training steps. This value tells us whether an increased activation generally benefits (negative average) or penalizes (positive average) the classification of the sample. Due to the use of float32 precision, zero partial derivatives appear at some point in training when the sample is correctly classified, making the gradient very small. Since the signs of these values are not relevant, they are ignored when the average sign is calculated.”

---

> ### Author Response · Authors · 2017-12-18
> **Part 2 of our answer**
>
> --------
> Fig 3: This figure studies “the first and last layers of each network”. Is the last layer really the last linear layer, the one followed by a softmax? In this case there is no relu and the 0 pre-activation is not meaningful (softmax is shift invariant). Or is the layer shown (e.g. “stage3layer2”) the penultimate layer? Minor: in this figure, it would be great if the plots could be labeled with which networks/datasets they are from.
> --------
>
> Penultimate, indeed. It was changed in the paper + we added dataset information to the plot captions
>
> --------
> Sec 5.2 states “neuron partitions the inputs in two distinct but overlapping categories of quasi equal size.” This experiment only shows that this is true in aggregate, not for specific neurons? I.e. the partition percent for each neuron could be sampled from U(45, 55) or from U(10, 90) and this experiment would not tell us which, correct? Perhaps this statement could be qualified.
> --------
>
> Indeed, some form of aggregation is done over neurons. The fact that a window centered around percentile rank 50 does not provide random predictions indicates that the percentile at which the two distributions cross each other changes across neurons, as explained in the paper: “While the partitions separate the inputs in equally sized categories on average, the size of the categories varies across neurons and is not exactly 50%, which explains the fact that a window center at the 50th percentile does not induce random predictions.”
> However, we keep some resolution about the position of the partition thresholds. In the example given in the comment, since “the performance should decrease when the window is located in fuzzy regions”, if the performance is lower for a (45-55) window, than for a (80-90) window, this means that the binary encoding is on average (across all neurons) more fuzzy in (45-55) window than in (80-90). It is thus more likely that the separation points between the categories are sampled from U(45,55), rather than from U(10,90).
> We hope that this answers the doubts of the reviewer, and are open to any further discussion.
>
> --------
> Table 1: “52th percentile vs actual 53 percentile shown”.
> --------
>
> Indeed. Thanks!
>
> --------
> > Table 1: The more fuzzy, the higher the percentile rank of the threshold
> This is true for the CIFAR net but the opposite is true for ResNet, right?
> --------
>
> Our statement is not based on the comparison of ReLU thresholds inside the same network (cifar CNN or ResNet), but across networks. Specifically, we have compared the ReLU thresholds for the penultimate layer in cifar CNN and in ResNet. In both networks, those thresholds correspond to similar percentile ranks for all neurons, indicating convergence to a precise value. Moreover, we observe that this percentile value is larger for ResNet than for the cifar CNN (84% vs 53%). Since ImageNet is a more complicated task, leading to fuzzier intermediate representations (see Figure 4), we state that “the more fuzzy, the higher the percentile rank of the threshold”. This statement is however not sufficiently supported by experimental evidences to lead to a definitive conclusion. We have decided to remove the ReLU analysis section (previously section 5.3), since it didn’t provide enough conclusive elements.

---

### Public Comment · (anonymous) · 2017-11-25
**You may discuss the relation to the recent NIPS publication https://arxiv.org/abs/1710.10328**

You may discuss the relation to the recent NIPS publication https://arxiv.org/abs/1710.10328.

Apparently your experimental findings can be explained in the lens of generalized hamming distance as introduced in the NIPS paper.  But I am not sure if there are anything more fundamental disclosed by your interesting experimental results.

---

> ### Author Response · Authors · 2017-12-18
> **We believe the relation with our work is too vague to be discussed.**
>
> Thank you very much for your interest in our paper.
> The cited paper shows that with a slight modification, the current operations in a neural network can be interpreted as a distance measure between fuzzy, binary variables from fuzzy logic theory. From this observation, the authors assume that a neural network treats activation and weights as fuzzy binary variables, without further verification.
>
> The observations of our paper show that a neural network treats activations as binary, fuzzy variables. However, we observe that this behavior emerges from training, and is not intrinsic to the operations of a neural network as –according to our understanding- is suggested by the cited paper and this anonymous comment. Further analysis of the cited paper shows that some important claims are not well validated. Indeed, a double thresholding activation function is introduced and motivated to outperform the ReLU activation. Validation of this claim (figure 4), however, compares the new activation function to linear activation (and not ReLU). Moreover, discussion of this figure compares convergence rates, which does not provide any relevant information, since the networks do not converge to the same final performance (84.63% accuracy for linear activation, 89.26% for double thresholding function).
>
> Overall, we believe the relation with Fan’s work is too vague to be discussed in our paper. We are however open to receive additional comments and clarifications about Fan’s interesting line of research.

---

### Decision · Program_Chairs · 2018-01-29
**ICLR 2018 Conference Acceptance Decision**

**Decision:**

Reject

**Comment:**

While one reviewer did upgrade their Rating from 6 to 7, the most negative reviewer maintains: "Overall, I find this work interesting and current results surprising. However, I find it to be a preliminary work and not yet ready for publication. The paper still lacks a conclusion / a leading hypothesis / an explanation for the shown results. I find this conclusion indispensable even for a small scientific study to be published." after the rebuttal. With scores of 7-5-4 it is just not possible for the AC to recommend acceptance.